# Regioselective Synthesis of 5-Trifluoromethyl 1,2,4-Triazoles via [3 + 2]-Cycloaddition of Nitrile Imines with CF_3_CN

**DOI:** 10.3390/molecules27196568

**Published:** 2022-10-04

**Authors:** Bo Lin, Zipeng Zhang, Yunfei Yao, Yi You, Zhiqiang Weng

**Affiliations:** 1Key Laboratory of Molecule Synthesis and Function Discovery, Fujian Provincial Key Laboratory of Electrochemical Energy Storage Materials, College of Chemistry, Fuzhou University, Fuzhou 350108, China; 2Fujian Engineering Research Center of New Chinese Lacquer Material, College of Materials and Chemical Engineering, Minjiang University, Fuzhou 350108, China

**Keywords:** trifluoromethyl, triazole, cycloaddition, nitrile imines, trifluoroacetonitrile

## Abstract

We herein describe a general approach to 5-trifluoromethyl 1,2,4-triazoles via the [3 + 2]-cycloaddition of nitrile imines generated in situ from hydrazonyl chloride with CF_3_CN, utilizing 2,2,2-trifluoroacetaldehyde *O*-(aryl)oxime as the precursor of trifluoroacetonitrile. Various functional groups, including alkyl-substituted hydrazonyl chloride, were tolerated during cycloaddition. Furthermore, the gram-scale synthesis and common downstream transformations proved the potential synthetic relevance of this developed methodology.

## 1. Introduction

1,2,4-Triazoles, five-membered heterocycles with three nitrogen atoms, prevalent in both natural and synthetic molecules, find use as drugs, insecticides, and synthetic materials. Their diverse biological activities in natural products and therapeutic agents include antifungal, antibacterial, antitubercular, antiviral, anti-inflammatory, anticancer, and analgesic activities [1,2,3,4]. Some potent molecules, such as deferasirox [5], 3-triazolylphenylsulfide [6], and sitagliptin [7], contain 1,2,4-triazole as the core structural framework (Figure 1).

Among these 1,2,4-triazoles, trifluoromethylated 1,2,4-triazoles have attracted significant attention [8]. It has been well established that the introduction of trifluoromethyl into therapeutic compounds can impart beneficial effects to lipophilicity, metabolic stability, conformational preference, and bioavailability [9,10,11]. As a consequence, a broad range of protocols for the construction of trifluoromethylated 1,2,4-triazoles have been disclosed [12,13,14,15,16,17,18,19,20,21,22,23,24,25,26,27,28].

However, current synthetic approaches for 5-trifluoromethyl 1,2,4-triazoles are scarce. Wu and co-workers reported a iodine-mediated annulation and a FeCl_3_-mediated cascade annulation of trifluoroacetimidoyl chlorides and hydrazones or hydrazides for the synthesis of 5-trifluoromethyl-1,2,4-triazoles (Figure 1a) [29,30]. Wu and co-workers subsequently developed a metal-free oxidative cyclization and a copper-catalyzed intramolecular decarbonylative cyclization reaction of trifluoroacetimidohydrazides with methylhetarenes or isatins for the synthesis of 5-trifluoromethyl-1,2,4-triazoles [31,32,33] and 2-(5-trifluoromethyl-1,2,4-triazol-3-yl)aniline derivatives [34], respectively. Darehkordi and co-workers described the synthesis of 1,3-diaryl-5-(trifluoromethyl)-1*H*-1,2,4-triazoles via the iodine-mediated intramolecular oxidative cyclization of *N*-(2,2,2-trifluoro-1-(arylimino)ethyl)benzimidamide intermediates, synthesized from the reaction of *N*-aryl-2,2,2-trifluoroacetimidoyl chlorides and benzamide hydrochloride derivatives (Figure 1c) [35]. More recently, the Ma research group developed a copper-catalyzed three-component reaction of aryldiazonium salts with fluorinated diazo reagents and nitriles, leading to two regiomers of trifluoromethylated N^1^-aryl-1,2,4-triazoles (Figure 1d) [36]. Although these previous reports were found to be effective, they suffered from some limitations, such as utilizing transition-metal catalysis, the use of the explosiveness of diazo and diazonium reagents, the complexity of the reaction mixture, and poor regioselectivities. Therefore, there is still a need for additional methodologies to access this important chemical scaffold from readily available starting materials under mild conditions.

As part of our ongoing interest centered on the synthesis of trifluoromethylated heterocycles [37], we recently disclosed a novel protocol for the construction of trifluoromethylated oxadiazoles from the reaction of nitrile oxide derivatives using 2,2,2-trifluoroacetaldehyde *O*-(aryl)oxime as the precursor of trifluoroacetonitrile [38]. Based on these results and considering that nitrile imines, generated in situ via the treatment of the hydrazonyl halides with stoichiometric amounts of base [39], could serve as the attractive C1N2 synthons in [3 + 2] cycloaddition reactions [40,41], we now investigated the feasibility of this approach for the regioselective synthesis of 5-trifluoromethyl 1,2,4-triazoles (Figure 1e).

## 2. Results

Our study commenced by choosing *N*-phenyl-benzohydrazonoyl chloride, **2a**, and trifluoroacetaldehyde *O*-(2,4-dinitrophenyl) oxime, **1**, as the model substrates (Table 1), using our previously reported optimized conditions for the synthesis of trifluoromethylated oxadiazoles [38]. To our delight, the reaction of **2a** with **1** in a 1:2 ratio in the presence of 2.0 equiv of NEt_3_ afforded the desired product, 5-trifluoromethyl 1,2,4-triazole, **3a**, in 3% and 8% yields in THF and DMSO, respectively (Entries 1 and 2). Among other solvents (Entries 3–6), the reaction worked best in CH_2_Cl_2_ (Entry 5), whereas in toluene, the reaction could not proceed (Entry 6). Notably, upon increasing the amount of **2a** and keeping the other parameters constant, the yield of **3a** was further improved (37% and 49% with **2a** and **1** in 1:1 and 2:1 ratios, respectively; Entries 7 and 8). Gratifyingly, the yield of **3a** was further improved to 75% by increasing the amount of NEt_3_ (3.0 equiv) (Entry 9). The best yield (83%) was obtained with the reaction of **2a** with **1** in a 1.5:1 ratio in the presence of 3.0 equiv of NEt_3_ in CH_2_Cl_2_ (Entry 10).

Subsequently, the efficacy of this developed protocol for the regioselective synthesis of 5-trifluoromethyl 1,2,4-triazoles with various hydrazonyl chlorides **2** was investigated (Figure 2). The reaction of 3-, or 4-Me-, and 4-*t*-Bu-substituted on the Ar moieties of hydrazonyl chlorides **2b**–**2d** with **1** provided annulated products **3b**–**3d** in 98%, 75%, and 60% yields, respectively. The variation in the electronic properties of the aryl substituents in the hydrazonyl chlorides had a significant effect on the reaction efficiency. The reaction of the hydrazonyl chloride possessing a strong electron-donating group such as -OMe (**2e**) successfully provided the corresponding triazole, **3e**, with good yield (75%). On the other hand, the substrates (**2f** and **2g**) containing strongly electron-withdrawing substituents (-CO_2_Me, -CF_3_, and –SO_2_N(*n*-Pr)_2_) furnished the desired triazoles (**3f**–**3h**) in moderate yields. These results are consistent with the previous observations in the synthesis of 1,2,4-triazol-3-ones by Wu and co-workers [42]. Of note, the halogen substituents, such as fluoro, and chloro, were also tested for the cycloaddition, and to our delight, good yields (56–74%) of the corresponding triazoles products, **3i**–**3l**, were obtained. A 2-naphthyl-substituted hydrazonyl chloride also furnished the desired product, **3m**, in 75% yield. Likewise, 2-thienyl substituted hydrazonyl chloride also smoothly underwent a reaction with **1**, giving the desired product, **3n**, in a 55% yield.

Meanwhile, cycloaddition was tested with different substituents at the Ar′ moieties of the hydrazonyl chlorides. Under the optimized conditions, a wide range of hydrazonyl chlorides contained different functional groups, such as methyl, *tert*-butyl, benzyloxy, trifluoromethyl, trifluoromethoxy, fluoro, chloro, and bromo groups, which were well tolerated and afforded triazole products **3o**–**3ab** in good yields (40–65%). 2-naphthyl-substituted hydrazonyl chloride was also efficient as a substrate to produce corresponding product **3ac** in a 67% yield. 

The structures of products **3a**–**3ac** were confirmed by the IR spectroscopy, and ^1^H, ^13^C{^1^H}, ^19^F NMR spectroscopy (Appendix A). Additionally, the structure of **3s** was unambiguously confirmed via single-crystal X-ray diffraction analyses (Figure 3).

Next, we attempted the cycloaddition of alkyl-substituted hydrazonyl chlorides **4** with **1** (Figure 4). Under the optimized conditions, both methyl, ethyl, and *n*-propyl-substituted hydrazonyl chlorides furnished the desired products, **5a**–**5c**, in only moderate yields (22–40%), which could be a result of the inherent instability of the hydrazonyl chloride substrates with an alkyl substituent. However, ester-substituted hydrazonyl chloride **4d** did not result in the desired cyclized product, **5d**, under our standard or modified reaction conditions, probably due to the resulting decreased nucleophilicity of the nitrogen atom through the conjugated system from the nitrile imine intermediates.

Nevertheless, a hydrazonyl chloride with a *t*-butyl substituent on the nitrogen atom reacted with **1** to produce the corresponding product, **5e**, in a 26% yield (Figure 2).

To assess the scale-up of the procedure, the reaction of **2a** with **1** as the representative example was investigated on a 10.0 mmol scale (Figure 3). Under the optimized reaction conditions, the cycloaddition occurred to give **3a** in a 56% (1.63 g) yield.

To further demonstrate the synthetic utility of the products, Heck reaction and Sonogashira coupling of bromide **3l** were carried out with *p*-methylstyrene and phenylacetylene to afford the corresponding products, **6** and **7**, in 62% and 71% yields, respectively (Figure 4a). Finally, the bromination of the C–H bond of **3e** was performed under oxidative conditions to furnish product **8** in a 74% yield (Figure 4b).

Based on the results obtained from these experiments and literature reports [38,43], a plausible mechanism for the formation of 5-trifluoromethyl 1,2,4-triazole (**3**) was proposed (Figure 5). The nitrile imine generated in situ from hydrazonyl chloride **2** in the presence of a base underwent regioselective [3 + 2] cycloaddition with the in situ-generated CF_3_CN from **1** to generate the desired product, **3**.

## 3. Materials and Methods

^1^H NMR, ^19^F NMR, and ^13^C NMR spectra were recorded using a Bruker AVIII 400 spectrometer. ^1^H NMR and ^13^C NMR chemical shifts were reported in parts per million (ppm) downfield from tetramethylsilane, and ^19^F NMR chemical shifts were determined relative to CFCl_3_ as the external standard; low field was positive. Coupling constants (*J*) are reported in Hertz (Hz). The residual solvent peak was used as an internal reference: ^1^H NMR (CDCl_3_ δ 7.26), ^13^C NMR (CDCl_3_ δ 77.0). The following abbreviations were used to explain the multiplicities: s = singlet, d = doublet, t = triplet, q = quartet, m = multiplet, br = broad. The infrared (IR) spectra were recorded using Nicolet iS50 at room temperature. HRMS were obtained from State Key Discipline Testing Center for Physical Chemistry of Fuzhou University. Trifluoroacetaldehyde *O*-(2,4-dinitrophenyl) oxime **1** [38], and *N′*-phenylacylhydrazides and hydrazonoyl chlorides [44,45] were prepared according to the published procedures. Starting materials and solvents that were received from commercial sources were used without further purification. Column chromatography purifications were performed via flash chromatography using Merck silica gel 60.

**Caution**: It is known that trifluoroacetonitrile is a highly toxic gas (boiling point, −64 °C) and must be handled with care. The rapid evolution of CF_3_CN gas occurs when this precursor reacts with base. All operations were performed in a fume hood under good conditions.

### General Procedure for the Synthesis of 5-Trifluoromethyl 1,2,4-Triazoles 3

A mixture of hydrazonoyl chloride, **2** (0.30 mmol, 1.5 equiv), and trifluoroacetaldehyde *O*-(2,4-dinitrophenyl) oxime, **1** (54.1 mg, 0.20 mmol, 1.0 equiv), in CH_2_Cl_2_ (1.0 mL) was added to a Schlenk tube equipped with a stir bar. Then, NEt_3_ (60.6 mg, 83.2 μL, 0.60 mmol, 3.0 equiv) was added. The tube was immediately sealed with a Teflon cap and stirred at room temperature for 12 h. After the reaction was terminated, the solvent was removed in vacuo under reduced pressure. Product **3** was purified via flash column chromatography on silica gel with petroleum ether and CH_2_Cl_2_ as eluent.

### Crystal Structure Analyses

The suitable crystals of **3s** were mounted on quartz fibers and X-ray data collected on a Bruker AXS APEX diffractometer, equipped with a CCD detector at −50 ºC, using MoKα radiation (λ 0.71073 Å). The data was corrected for Lorentz and polarisation effect with the SMART suite of programs and for absorption effects with SADABS [46]. Structure solution and refinement were carried out with the SHELXTL suite of programs. The structure was solved by direct methods to locate the heavy atoms, followed by difference maps for the light non-hydrogen atoms. CCDC 2183507 contain the supplementary crystallographic data. These data can also be obtained free of charge at ccdc.cam.ac.uk/structures/ from the Cambridge Crystallographic Data Centre.

## 4. Conclusions

In summary, we accomplished a [3 + 2]-cycloaddition of nitrile imines with CF_3_CN for the synthesis of 5-trifluoromethyl 1,2,4-triazoles utilizing 2,2,2-trifluoroacetaldehyde *O*-(aryl)oxime as the precursor of trifluoroacetonitrile. The exclusive regioselectivity, functional group tolerance, mild reaction conditions, and efficient scalability are the important practical advantages.

## Data Availability

The data presented in this study are available in this article.

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
