# Peer review of "Regioselective Synthesis of 5-Trifluoromethyl 1,2,4-Triazoles via [3 + 2]-Cycloaddition of Nitrile Imines with CF3CN"

_molecules, 2022, doi:10.3390/molecules27196568_

Round 1

Reviewer 1 Report

This manuscript describes regioselective synthesis of 5-trfluoromethyl 1,2,4-trazoles.  It is well written, and the new method would be useful for synthesis of these kinds of compounds.  Therefore, I think this manuscript could be acceptable as an original paper in Molecules after considering the following comments.

I think 4.0 mmol scale is not so big scale.  I hope to try more big scale about more than 10 mmol, if possible.

Reaction mechanism on generation of nitrile imine from hydrazonyl chloride is unique.  I think the authors should draw the reaction mechanism on generation of nitrile imine from 3 by curved arrow.

Minor.

Abstract; change the style of O to italic of O-(aryl)oxime.

Scheme 1; it is hard for me to see the structures by gray.

Figure 2; what is two dots?

Scheme 5; change the style of 3 to bold.

Author Response

Question/Comment 1:  I think 4.0 mmol scale is not so big scale.  I hope to try more big scale about more than 10 mmol, if possible.

Response:  We sincerely thank the reviewer for his/her good suggestion. The scale of reaction was increased up to 10 mmol level, and these results have been added in the revised manuscript.

Question/Comment 2:  Reaction mechanism on generation of nitrile imine from hydrazonyl chloride is unique.  I think the authors should draw the reaction mechanism on generation of nitrile imine from 3 by curved arrow.

Response: We have revised the reaction mechanism in Scheme 5.

Minor.

Question/Comment 3: Abstract; change the style of O to italic of O-(aryl)oxime.

Response: The style of O of O-(aryl)oxime has been revised to italic.

Question/Comment 4: Scheme 1; it is hard for me to see the structures by gray.

Response: The structures in Scheme 1 have been revised by black.

Question/Comment 5: Figure 2; what is two dots?

Response: The Figure 2 has been re-plotted.

Question/Comment 6: Scheme 5; change the style of 3 to bold.

Response: The style of 3 has been revised to bold

Reviewer 2 Report

In this manuscript, the authors report the new efficient [3+2] cycloaddition method to give regioselective trifluoromethylated triazoles under very practical conditions. This is a very nice synthetic extension of the authors’ previous study for utilization of 2,2,2-trifluoroacetaldehyde O-(aryl)oxime as a trifluoroacetonitrile precursor (see ref. 38). The thus generated trifluoroacetonitrile can be utilized as a useful counterpart for nitrile imines in regioselective [3+2] cycloadditions. The room-temperature reaction is practical, and gram-scale application is possible without significant loss of the reaction efficiency. The scope and advantages of the reactions seem to be in accord with required standard of the synthetic work published in Molecules journal. The experimental details are well documented in SI with full spectroscopic data for compound characterizations. Accordingly, this reviewer would recommend provisional acceptance of this work including the nice synthetic development in Molecules journal, while there need several minor revisions to enhance the quality of the manuscript;

1) It is not clear that the product yields are calculated on the starting material 1 basis.

2) In Scheme 5, the used triethylamine is not an anionic base.

3) The author should clarify in the manuscript the controlling factor of the regioselectivity during the reaction (themodynamical or kinetical).

Author Response

Question/Comment 1: It is not clear that the product yields are calculated on the starting material basis.

Response: Thank you for your valuable comments and suggestions. We have addressed this in the revised manuscript in the footnote in Table 2 and 3 as “(calculated on starting material basis).”

Question/Comment 2: In Scheme 5, the used triethylamine is not an anionic base.

Response: This has been corrected in the revised manuscript.

Question/Comment 3: The author should clarify in the manuscript the controlling factor of the regioselectivity during the reaction (themodynamical or kinetical).

Response: In this reaction, 5-trifluoromethyl 1,2,4-triazoles were generated through [3 + 2]-cycloaddition of nitrile imines with CF3CN (a highly electrophilic reagent) The precursor 1 rapidly releases CF3CN gas in the presence of base and the nitrile imine intermediate is generated from hydrazonyl chlorides under the basic conditions. The 1,3-dipolar cycloaddition reaction of CF3CN and nitrile imine was occurred to furnish the desired 5-trifluoromethyl 1,2,4-triazole products. This cycloaddition reaction follows the electrical law. The nitrogen anion of nitrile imide attacked carbon atom of CF3CN. Meanwhile the nitrogen atom of CF3CN becomes nitrogen anion to attack the carbon atom of nitrile imine, which is resulted in the formation of triazole products. This has been elearly shown in Scheme 5.

Reviewer 3 Report

This manuscript describes the full details of the regioselective synthesis of trisubstituted triazole derivatives having a trifluoromethyl group.  As authors mentioned, the importance of this type of compounds are well recognized with their biological activities.  Although the yields of the aliphatic substituted derivatives ate relatively low compared to the aromatic substituted one, the synthetic utility and relatively simple synthetic procedure make the value of the current work.

Chemistries described in this paper would be a nice piece of work and of interest for a number of researchers in this field.

I recommend this manuscript for publication in Molecules.

Author Response

Question/Comment: This manuscript describes the full details of the regioselective synthesis of trisubstituted triazole derivatives having a trifluoromethyl group.  As authors mentioned, the importance of this type of compounds are well recognized with their biological activities.  Although the yields of the aliphatic substituted derivatives ate relatively low compared to the aromatic substituted one, the synthetic utility and relatively simple synthetic procedure make the value of the current work.

Chemistries described in this paper would be a nice piece of work and of interest for a number of researchers in this field.

I recommend this manuscript for publication in Molecules.

Response: We appreciate the comments of the reviewer. There is no question from this reviewer.